# Immune Responses Elicited by Live Attenuated Influenza Vaccines as Correlates of Universal Protection against Influenza Viruses

**DOI:** 10.3390/vaccines9040353

**Published:** 2021-04-07

**Authors:** Yo Han Jang, Baik L. Seong

**Affiliations:** 1Department of Biological Sciences and Biotechnology Major in Bio-Vaccine Engineering, Andong National University, Andong 1375, Korea; yhjh0323@anu.ac.kr; 2Vaccine Industry Research Institute, Andong National University, Andong 1375, Korea; 3Department of Biotechnology, College of Life Science and Biotechnology, Yonsei University, Seoul 03722, Korea; 4Vaccine Innovation Technology Alliance (VITAL)-Korea, Yonsei University, Seoul 03722, Korea

**Keywords:** influenza live attenuated vaccine, universal vaccine, antibody, T cell, correlate of protection

## Abstract

Influenza virus infection remains a major public health challenge, causing significant morbidity and mortality by annual epidemics and intermittent pandemics. Although current seasonal influenza vaccines provide efficient protection, antigenic changes of the viruses often significantly compromise the protection efficacy of vaccines, rendering most populations vulnerable to the viral infection. Considerable efforts have been made to develop a universal influenza vaccine (UIV) able to confer long-lasting and broad protection. Recent studies have characterized multiple immune correlates required for providing broad protection against influenza viruses, including neutralizing antibodies, non-neutralizing antibodies, antibody effector functions, T cell responses, and mucosal immunity. To induce broadly protective immune responses by vaccination, various strategies using live attenuated influenza vaccines (LAIVs) and novel vaccine platforms are under investigation. Despite superior cross-protection ability, very little attention has been paid to LAIVs for the development of UIV. This review focuses on immune responses induced by LAIVs, with special emphasis placed on the breadth and the potency of individual immune correlates. The promising prospect of LAIVs to serve as an attractive and reliable vaccine platforms for a UIV is also discussed. Several important issues that should be addressed with respect to the use of LAIVs as UIV are also reviewed.

## 1. Introduction

Influenza viruses have posed serious threats on human public health worldwide despite development of effective vaccines and antiviral drugs. Each annual influenza epidemic affects 5−15% of the population and causes 3−5 million cases of hospitalization, claiming 290,000−650,000 lives worldwide [1]. The substantial levels of antigenic diversity and variability of influenza viruses and zoonotic transmission of non-human influenza viruses to humans present persistent possibilities of human infections with novel influenza viruses to which most population has little or no preexisting immunity. Currently-licensed seasonal influenza vaccines have proven effective against well-matched strains. There are three types of influenza vaccines clinically used for humans; inactivated influenza vaccines (IIVs), live attenuated influenza vaccines (LAIVs), and recombinant hemagglutinin (HA) subunit vaccines. IIVs and HA subunit vaccines primarily induce HA-specific neutralizing antibodies that inhibit the binding of viral HAs to cellular receptor sialic acids, thereby preventing viral entry into cells. HA inhibitory antibodies, however, provide very narrow strain-specific protection since the HA head region harboring the receptor binding site is highly variable among influenza viruses. Therefore, antigenic change in the HA head region by antigenic drift often leads to viral escape from the antibodies. This is why the HA and NA of seasonal influenza vaccines are updated almost annually to match newly-circulating strains. In addition, pandemic outbreaks often result from genetic reassortment between influenza viruses from different species, which is unpredictable when a pandemic will occur. In the case of a pandemic, most of the population remains vulnerable to infection with the novel pandemic strain until a well-matched vaccine becomes available.

Extensive efforts to develop a universal influenza vaccine (UIV) that provides broad protection against diverse influenza viruses have been made worldwide [2]. Induction of antibodies and cell-mediated immune responses directed to conserved viral antigens is the key to eliciting broad protection. Since the discovery of broadly neutralizing antibodies to the conserved HA stalk region, several strategies have been advanced, such as chimeric HAs and headless HAs. The HA stalk-based approaches have been successful to selectively induce HA stalk antibodies that exhibit broad protection in animal models and are currently under clinical evaluation. In addition to HA stalk-based approaches, a number of rational strategies have been designed to express cross-reactive antigens (such as NA or M2e) or T cell epitopes in multiple vaccine platforms, such as viral vectored vaccine, recombinant protein or peptide vaccines, DNA or RNA vaccines, virus-like particles, and nanoparticles [3]. LAIVs have shown superior protection efficacy not only against homologous influenza viruses but also against mismatched heterologous strains. In particular, cell-mediated immune responses elicited by LAIVs are considered as critical for cross-protection, and other factors, such as mucosal IgA antibodies and non-specific protection, have also been shown to correlate with cross-protection. Despite abundant experimental and clinical evidence for cross-protection, LAIVs have received little attention as target platforms for a UIV, with only a few recent studies demonstrating exceptional cross-protection abilities by LAIVs in animal models. Studies have determined that induction of the multiple correlates of protection are necessary for providing broad and potent cross-protection against both HA group 1 and 2 influenza A viruses [4]. It has been suggested that antibody effector functions, NA antibodies, and mucosal IgA antibodies are important for cross-protection and thus should be included in a UIV. This review discusses the potential of LAIVs to serve as a reliable UIV platform, giving special focus on the breath and potency of individual immune responses elicited by LAIVs. Several important considerations on developing LAIV-based UIVs, such as low efficacy in elderly people, preexisting immunity, and safety issues, are also discussed.

## 2. Principles and Cross-Protection of LAIVs

### 2.1. Cold-Adapted Live Attenuated Influenza Vaccines

Among various types of LAIVs developed so far, cold-adapted LAIVs (CAIVs) are currently licensed for clinical uses in humans. CAIVs are established by serial passages of parental influenza viruses at low temperatures in embryonated chicken eggs or chicken cells, making the virus less replicative at normal and elevated body temperatures. The resulting cold-adapted virus exhibits cold-adapted (ca), temperature-sensitive (ts), and attenuated (att) phenotypes and, thus, can be used as a safe CAIV donor strain. Genetic and phenotypic analysis have revealed that mutations in the polymerase genes and the NP gene are crucial for the expression of the ca, ts, and att phenotypes [5,6,7,8]. A particular CAIV is constructed by genetic reassortment with the six internal cold-adapted donor strains and the two surface genes, HA and NA, originated from a wild-type circulating virus. Five strains of CAIVs are currently licensed for the manufacture of seasonal CAIVs: A/Leningrad/134/17/57 (H2N2), A/Leningrad/134/47/57 (H2N2), A/Ann Arbor/6/60 (H2N2), B/USSR/60/69, and B/Ann Arbor/1/66 strains. These CAIVs have served as donor strains for the development of seasonal influenza A and B virus vaccines and also other pandemic or prepandemic vaccines [9]. While CAIVs based on A/Leningrad/134/17 and B/USSR/60/69 strains are used in people aged three or older, CAIVs based on A/Ann Arbor/6/60 and B/Ann Arbor/1/66 strains are licensed for use for people aged 2–49 years. Other independent CAIVs, A/X-31 (H3N2) and B/Lee/40 ca, were generated in South Korea but have not entered clinical development yet. CAIVs are delivered intranasally, via the same route of entry as wild-type influenza viruses and, thus, are expected to induce similar immune responses to those by natural infection. CAIVs only weakly replicate in the upper respiratory tracts but rarely in the lower respiratory tracts at normal or increased body temperatures. Replication of CAIVs induce antibody responses and cell-mediated immune responses both systemically and locally, generating multiple immune arms for protection (Figure 1). The immune responses elicited by a CAIV include local mucosal IgA antibodies, serum hemagglutinin-inhibitory (HI) and neutralizing antibodies, and T cell immunity [10]. While it is generally believed that these multiple factors cooperatively contribute to superior protection as compared to IIV, particular functional roles of individual factors have not been clearly defined yet. 

### 2.2. Other Types of LAIVs

Reverse genetics technology has made it possible to introduce targeted mutations into the genome of influenza viruses for generation of a number of genetically engineered influenza viruses. This resulted in various attenuation strategies for the development of novel types of LAIVs (Table 1). Influenza NS1 is a non-structural protein that acts as an interferon antagonist, and deletion or truncation of the NS1 has been shown to result in significant attenuation of viral replication. Studies have shown that monovalent or multivalent LAIVs expressing truncated or mutated NS1 are highly attenuated, immunogenic, and provide robust protection against homologous as well as heterologous influenza virus infections in animals and humans [11]. Of note, a recent study showed that del-NS1 H1N1 LAIV provided broad protection against H1N1, H5N1, and H7N9 challenges in mice through cross-reactive T cell responses even in the absence of neutralizing antibodies [12]. Furthermore, introduction of caspase recognition sites into the NP and NS1 proteins of an influenza virus resulted in caspase-dependent cleavage of the proteins in virus-infected cells, leading to significant attenuation of the viruses [13]. The mutant viruses were highly attenuated and immunogenic in mice, providing strong protection against homologous challenge. Modification of HA cleavage site into elastase recognition motif resulted in the generation of an attenuated influenza virus that could replicate only in the presence of elastase in cell culture [14]. Subsequent follow-up studies examined the potential of the elastase-dependent influenza viruses to serve as novel LAIVs in animal models [15,16]. A double-attenuated LAIV with elastase-susceptible HA cleavage site and shortened NS1 protein demonstrated increased safety but was still immunogenic in a swine model [15]. Similar strategy could be extended to LAIV against influenza B virus [16]. Introduction of miRNA targeting sequences into NP gene resulted in viral gene silencing and species-specific attenuation of the viruses [17]. miR-21- or miR-192 targeted influenza viruses were highly attenuated in susceptible cells and mice but provided robust homologous and heterologous protection in mice [18,19]. M2-deficient influenza virus infected cells only a single round but could induce robust antibody responses and T cell responses providing effective heterosubtypic protection in mice and ferrets [20,21]. Additionally, genetic engineering of splicing elements in M gene and NS gene and codon deoptimization of NS gene also attenuated influenza viruses, providing additional options for developing LAIVs [22,23]. It should also be noted that over-attenuation of the virus may limit vaccine productivity and also the immunogenicity of LAIVs. Extensive evaluation for the productivity, safety, immunogenicity, and potential risk of reversion into a virulent strain is needed for further development into clinically-relevant LAIV options. 

### 2.3. Cross-Protection by LAIVs

It is well-documented that natural infection with an influenza virus induces broadly reactive T cell responses that provide cross-protection not only against homologous but also against drifted strains and even antigenically distant viruses with different subtypes [24]. The same mechanisms may hold true for cross-protection elicited by LAIVs, although the magnitude of T cell responses can be milder than natural infection due to highly restricted replication of the attenuated vaccines. In the literature, there is a large body of experimental evidence for cross-protection by LAIVs [10]. LAIVs have demonstrated considerably varying levels of protection window in different studies from subtype-specific to pan-influenza A protection. A limited number of controlled human studies showed cross-protection of CAIVs against drifted H1N1 or H3N2 strains that were not contained in the vaccines [25,26]. In animal studies, LAIVs have been shown to provide heterosubtypic and even heterotypic protection [10]. A/Ann Arbor/6/60 ca (H2N2) donor strain and A/Ann Arbor/6/60 ca-based 2009 pandemic H1N1 vaccine provided heterosubtypic cross-protection against H5N1 viruses in mice and ferrets [27,28]. A/X-31 ca (H3N2) donor strain protected mice from H1N1 challenge [29], and A/X-31 ca-based 2009 pandemic H1N1 vaccine provided protection against H1N1, H3N2, and H5N2 viruses in mice, showing broad protection against both HA group 1 and 2 influenza A viruses [30]. Of note, the A/X-31 ca donor strain provided heterotypic protection against influenza B virus in a mouse model, although the protection was shown to be mediated by innate immunity not by specific antibody responses. While cross-protection by LAIVs has been extensively examined in animal models, the potency and the breadth of cross-protection in humans remain poorly understood due to several limitations. In humans, controlled challenges with antigenically distant viruses to a vaccine strain is hardly possible due to potential risk of lethal infection even in vaccinated groups. Moreover, in vitro measurements of antibody or T cell-mediated protection against heterologous viruses cannot be interpreted as genuine protection efficacy in vivo since in vitro assays reflect protection mediated by only a part of whole immune correlates induced by vaccination. As discussed below, cross-protection by LAIVs can be ascribed to multiple correlates in addition to T cell responses (Table 2). Better understanding on protection mechanisms by individual immune correlates against influenza viruses and deeper characterization of immune responses induced by LAIVs present the first step towards a development of a truly UIV based on a LAIV. 

## 3. Broadly Protective Antibody Responses Elicited by LAIVs

### 3.1. HA Stalk Antibody Responses

Induction of broadly neutralizing antibodies (bnAbs) specific to conserved HA stalk region have been considered as the key to the development of a UIV. To date, a number of antibodies targeting the HA stalk capable of neutralizing a wide range of influenza strains and subtypes have been characterized [31]. HA stalk bnAbs have been isolated from animals (mostly mice) and humans with prior influenza infections or vaccinations. Those HA stalk bnAbs can be divided into two classes according to their breadth of neutralizing activities; HA group-specific and pan-influenza A. HA group-specific bnAbs can neutralize either group 1 or group 2 influenza A viruses, whereas pan-influenza A bnAbs can neutralize both groups. Notably, a study showed that genetically engineered multi-domain antibody MD3060 neutralized both influenza A and B viruses [32]. The neutralizing ability of HA stalk bnAbs include multiple independent mechanisms. Many of bnAbs bind to and stabilize HAs in its prefusion state or inhibit proteolytic processing of HA0 precursors into HA1 and HA2 subunits, preventing pH-dependent conformational change of the protein required for membrane fusion between virus and endosome [33]. Other HA stalk bnAbs prevent virus particle release from cells or inhibit NA activity by steric hindrance [34,35]. It is likely that slightly different epitopes in HA stalk or binding angle to the corresponding epitopes may explain the differences in neutralizing potency and breadth of individual HA stalk bnAbs [34,36,37]. In addition to neutralizing activities, antibody effector functions have been suggested as crucial correlates of protection by HA stalk bnAbs. Three types of antibody effector functions are observed by HA stalk bnAbs; antibody-dependent cellular cytotoxicity (ADCC), antibody-dependent cellular phagocytosis (ADCP), and complement-dependent cytotoxicity (CDC), which mediate elimination of viruses or virus-infected cells. Studies have demonstrated that antibody effector functions of HA stalk bnAbs have comparable and sometimes more important roles for eliciting broad protection than their neutralizing activities. For instance, in vitro neutralizing activity of HA stalk bnAb 6F12 disappeared when the interaction between the Fc and Fc receptor (FcR) was blocked by Fc replacement or mutation in a mouse model, suggesting that in vivo protection by the antibody can be mainly ascribed to effector functions not to neutralizing activity [38]. Other HA stalk bnAbs FI6 and 2G02 also showed a similar protection mode to the 6F12 antibody [39]. While those antibodies demonstrated potent in vitro neutralizing activity against the 2009 pdmH1N1 virus, mutant antibodies lacking binding ability to FcR rarely protected mice from infection with the same strain, showing that in vitro neutralizing activity minimally contributed to protection in vivo. 

Several studies have shown that LAIVs also induce HA stalk antibodies in animals and humans. A mouse study showed that prime-boost vaccination with LAIVs each carrying different subtypes of full-length HA proteins induced robust HA stalk antibodies [40]. Although LAIV-induced antibodies did not show in vitro viral neutralizing activity, the antibodies demonstrated cross-reactive ADCC activities against four different H1N1, H3N2, H5N2, and H7N1 strains and contributed to broad protection. Considering that LAIVs induce polyclonal antibody responses to multiple influenza viral proteins, it is likely that ADCC activities of antibodies induced by LAIVs represent the cumulative effects of polyclonal antibodies mixture. However, a ferret study showed that three doses of LAIVs carrying full-length and non-chimeric Has, including H5, H8, and H9, rarely induced HA stalk antibodies, suggesting that HA stalk antibodies were poorly induced by boosting with different HA stalks [41]. Interestingly, no significant differences in in vitro neutralizing activities of serum antibodies and in vivo protection against the challenge were observed between cHA-based LAIVs and classical LAIVs. The disparity of results of HA stalk antibody induction by LAIVs between mice and ferrets may result from differences in LAIV strains used, animal species, or experimental settings. Whether HA stalk antibodies are induced by LAIVs in humans was also investigated. A study showed that trivalent LAIVs elicited H3-head and low levels of H1 stalk antibody responses in children [42]. In that study, it was suggested that HA stalk antibodies could be effectively boosted by LAIVs when children have preexisting HA stalk-directed immunity. By contrast, LAIVs dominantly boosted H3 head antibodies to stalk antibodies in children who have low levels of preexisting immunity to H3N2 strain. This result suggests that preexisting immunity to the HA stalk can be beneficial for the induction of HA stalk antibodies by LAIVs in humans. The following study investigated the functionality of HA stalk antibodies after LAIV in children and adults [43], in which LAIV-induced HA stalk antibodies exhibited viral neutralizing and ADCC activity, which was more significant in children than adults. While it remains unclear whether HA stalk antibodies induced by LAIVs contribute to protection, animal and human studies together demonstrate that LAIVs are able to induce HA stalk antibody responses, providing important implications on developing LAIV-based UIVs. Several studies used LAIVs as the delivery vehicle of the HA stalk through genetic engineering to carry cHAs. Multiple administrations of cHAs by recombinant influenza virus, LAIV, and IIV-induced HA stalk antibodies systemically and locally in ferrets, providing excellent protection against the 2009 pandemic challenge [44]. Subsequent study investigated the protective efficacy of cHA-based LAIV-LAIV, LAIV-IIV, and one dose of LAIV regimens in ferrets, demonstrating that both one and two doses of LAIVs, and also LAIV-IIV induced HA stalk antibody responses [45]. Those results together suggest that LAIVs can induce broadly protective HA stalk antibodies in animals and humans, providing a promising prospect of using LAIVs as a UIV platform.

### 3.2. M2e Antibodies

The M2 protein is the third most abundant surface protein of an influenza virus. M2 is also expressed on the surface of virus-infected cells, generating antibodies to the extracellular domain of M2 (M2e). In the studies of LAIVs reported thus far, protective roles of M2e antibodies are not the main focus since it is believed that M2e is less abundant and M2e antibodies contribute less to protection than antibodies to the surface HA and NA proteins. Several strategies have been designed to increase M2e antibody responses, including VLP, DNA, peptide, and protein vaccines [46]. M2e is highly conserved among influenza viruses and thus has been an attractive target for developing a UIV. While M2e antibodies are able to restrict in vitro replication of multiple strains of influenza A viruses [47], it has been shown that the main protective roles of M2e antibodies in vivo are antibody effector functions of ADCC, ADCP, and CDC, rather than direct viral neutralization [46]. Although there is no direct report addressing M2e antibody-dependent protection elicited by LAIVs, induction of M2e antibodies by LAIVs can be reasonably inferred from reported induction of M2e antibody responses following influenza virus infection. It was shown that influenza infection induced weak anti-M2e antibody responses for a short period in animals and humans [48]. However, M2e antibody responses can be boosted by subsequent reinfections by heterologous influenza viruses. Although primary infection of pigs with the high doses of H1N1 virus induced very low levels of M2e antibodies, reinfection with H3N2 virus resulted in a 25-fold increase in the M2e antibody titers [49]. Likewise, mice sequentially infected with PR8 (H1N1) and X-31 (H3N2) strains induced detectable M2e antibodies [50]. Those results indicate that poor M2e antibodies elicited by primary infection can be boosted by subsequent reinfections with homologous or heterologous strains. Considering that many people are exposed to influenza infections during their lifetime and, therefore, have preexisting immunity directed to M2e, it is likely that one or two doses of LAIVs boost M2e antibody responses. Consistent with this speculation, a study demonstrated that seroprevalence of M2 antibodies increased with age in humans and that infection with 2009 pandemic strain recalled M2e antibody responses, with more robust boosting observed in individuals who had preexisting immunity to M2e [51]. The report also showed that M2e antibodies induced by infection was cross-reactive to M2e from unrelated seasonal influenza A viruses. The results present the possibility that LAIVs recall M2e antibody responses in humans with prior exposures to influenza viruses. Further studies are needed to confirm the induction of M2e antibodies and their protective roles following LAIVs. 

### 3.3. NA Antibodies

NA is the second most abundant surface glycoprotein that has receptor-destroying enzyme activity. The NA cleaves terminal sialic acid reside bound to viral HA and releases progeny virus particles from infected cells, facilitating the virus spread in a host. NA has long been ignored as vaccine antigens since the NA is less abundant than HA on the virion and NA antibodies do not prevent viral entry into cells. However, recent findings on the diverse roles of NA during virus infection cycle underpin the importance of NA antibodies for protection against influenza viruses. In addition to the classical supportive role for viral egress at the later stage of infection, multiple functions of NA have also been demonstrated. It was shown that the sialidase activity of NA is also important for viral entry at the early stage of infection. In the mucus, viral HAs of influenza virus are captured by mucin proteins that are highly glycosylated, and NA cleaves the sialic acid from the mucins and releases the captured viral particles, permitting the viruses to reach host cells [52]. It was also shown that removal of the sialic acids from viral HA by NA helps HA-dependent membrane fusion and enhances viral infectivity [53]. Surprisingly, a couple of studies showed that some of H3N2 viruses use NAs for receptor binding instead of HA, demonstrating the novel role of NA during viral entry step [54]. Those results together demonstrate that NA has multiple roles during entire infection cycle, implying that NA antibodies may make a considerable contribution to protection against influenza viruses. Recently, a number of broadly protective monoclonal NA antibodies were identified from animals and humans. Similar to HA stalk antibodies, multiple protection mechanisms of NA antibodies were characterized in animal models, including inhibition of the enzyme activity of NA (NI activity), antibody effector functions (ADCC, ADCP, and CDC), and viral neutralizing activity [55]. A growing number of animal and human studies has suggested that NA antibodies are important and independent correlates of protection, in addition to HI antibodies and HA stalk antibodies. The protection breadth of NA antibodies substantially varies from strain-specific to pan-influenza depending on their corresponding epitopes, and the cross-protective ability of NA antibodies were evaluated by several NA-based vaccine constructs in animal models [56]. Several recent studies have demonstrated the protective roles of NA antibodies elicited by LAIVs in humans. Seasonal LAIV induced NA-inhibitory (NI) antibodies not only to homologous H1N1 strain but also to heterologous H1N1 and H5N1 strains, demonstrating the cross-reactivity of NA antibodies after vaccination with a LAIV in humans [57]. The study showed no correlation between NI and HI antibody titers, suggesting that independent detection of HI antibodies and NI antibodies provide a more accurate assessment of LAIV immunogenicity. Another study demonstrated that LAIV resulted in significant increases in NI antibody titers in most of children and adults in the presence of preexisting NI antibodies [43]. Animal studies also demonstrated robust induction of cross-reactive NA antibodies following LAIVs. Prime-boost LAIVs carrying only N1 subtype induced exceptional breadth of cross-reactive NA antibodies covering N1, N2, N6, N7, N8, and N9 in a mouse model [40]. Furthermore, cHA vaccine regimens involving LAIV induced detectable level of NA antibodies in ferrets, in which NA antibodies were suggested as correlate of protection, in addition to HA stalk antibodies [44]. One of the benefits of using LAIVs is the delivery of NA antigens in their native confirmation. A recent study showed that influenza infection in humans induced broadly protective NA antibodies (23%) that was comparable to HA antibodies (35%), whereas IIV induced predominantly HA antibodies (87%) and very low level of NA antibodies (1%) [58]. This study suggested that insufficient content and incorrectly folded NA antigens is the most likely reason for low NA antigenicity of IIV and also proposed that LAIVs would be a better option for eliciting broadly protective and functional NA antibodies. 

### 3.4. Antibodies to Internal Viral Proteins

In addition to surface proteins HA, NA, and M2, influenza infection induces antibodies directed to internal structural NP and M1, nonstructural polymerase complexes, NS1, and NEP, and even small accessory proteins such as PB1-F2 and PA-X [59,60,61,62]. Since the internal proteins are not exposed on the surface on a native virion, it is likely that internal proteins released from degraded virus particles or infected cells or expressed on infected cell membrane mediate antibody generation. The magnitude, quality, and protective functions of antibodies to the influenza internal proteins have not been well-defined yet. Given the highly conserved nature of the internal antigens, however, it is likely that antibodies to the internal proteins have broadly protective roles against diverse influenza strains. The protective role of NP antibodies was investigated in animal challenge models. Mice immunized with recombinant NP proteins demonstrated significantly reduced morbidity upon heterologous influenza infections in an antibody Fc-dependent manner [63,64]. Although it has been shown that NP can be expressed on cell membrane of virus-infected cells [65], whether NP antibodies exert antibody effector functions remains controversial. A study showed that human NP and M1 antibodies were capable of activating NK cells but did not kill target cells in vitro [66]. By contrast, another study showed that human NP antibodies had ADCC activity against diverse strains including H1N1, H3N2, and H7N9, suggesting broad reactivity of ADCC-inducing NP antibodies [67]. Human NP antibodies demonstrated no detectable CDC activity [68], whereas murine NP antibodies showed low but detectable CDC activity [69], showing contrasting results depending on species or experimental settings. Given that NP and M1 are highly conserved among influenza A viruses, it would be worthwhile to determine the potency and breadth of protective roles of the corresponding antibodies. A study demonstrated interesting results regarding the protective role of PB1-F2 antibodies generated during influenza virus infection [70]. While antibodies specific to the N-terminal of PB1-F2 had no protective effects, antibodies to the C-terminal of the protein partially protected mice against influenza infection. It was assumed that antibodies to the N-terminal part of the PB1-F2 dominated over antibodies to the C-terminal part of the protein as protective effects were lost when both antibodies were present. To date, however, there is no report addressing the protective functions of antibodies directed to other internal proteins such as polymerase complexes, NS1, NEP, and small accessory proteins. 

## 4. Mucosal Immunity

The mucosal surface of respiratory tract is the entry site of influenza viruses and, thus, anti-influenza immunity in this site is very important for preventing the viral infection. LAIVs are administered into the nasal route and primarily infect the upper respiratory tract and stimulate mucosal immune responses. There are distinctions in antibody types and antibody titers between the upper and lower respiratory tracts. The lower respiratory tract contains higher titers of IgG antibodies than IgA antibodies with a ratio of 2.5:1, whereas the upper respiratory tract is dominated by IgA antibodies with an IgG to IgA ratio of 1:3 [71]. It is assumed that while mucosal IgA antibodies have the same binding specificity to IgG antibodies, IgA antibodies exhibit broader reactivity than IgG antibodies due to increased avidity to antigens by their propensity to form multimeric structure [72]. Whereas systemic IgG antibody-mediated protection against influenza virus is well-defined, only few data are available for mucosal IgA antibody-dependent protection, especially with regard to cross-protection. It was initially reported that mucosal IgA antibodies were more correlated with cross-protection against heterologous influenza viruses than serum antibodies or cytotoxic T cells in mice [73]. Of note, mucosal IgA antibodies to HA demonstrate binding capacity and neutralizing activity against both homologous and heterologous strains, demonstrating functional protective effects of IgA antibodies [74,75]. Those observations of cross-protection ability of mucosal IgA antibodies were extended into mucosal vaccination with CTB adjuvant for broad protection against heterologous influenza viruses. Intranasal immunization with HA proteins and IIVs induced robust cross-reactive mucosal IgA antibodies, providing superior cross-protection to parenteral vaccination [76,77]. Cross-protection by mucosal IgA antibodies was also confirmed in knock-out mice in which transepithelial transport of polymeric IgA antibodies was blocked [78]. When formalin-inactivated IIV was intranasally administered, mice were protected from heterologous influenza strains despite the lack of neutralizing activity of mucosal IgA antibodies, suggesting that non-neutralizing IgA antibodies may be responsible for cross-protection [79]. Several adjuvants were suggested to enhance mucosal immunity of influenza vaccines, including CTB, chitin microparticles, poly(I:C) dsRNA, polyI:polyC_12_U, intatolimod (TLR-3 agonist), and surf clam microparticles, and also novel immune stimulating materials, such as mushroom mycelia extracts and alpha-galactosyl ceramide [80,81,82,83,84]. Overall, those studies showed that intranasal administration of diverse influenza vaccine types including IIVs, proteins vaccines, DNA vaccines, viral vectored vaccines conferred broad heterotypic protection by inducing mucosal IgA antibodies. Protection mechanisms of IgA antibodies include both neutralizing and non-neutralizing activities. It has been previously demonstrated that polymeric IgA antibodies specific to HA neutralize multiple influenza A strains more effectively than monomeric IgA or IgG antibodies [85,86], suggesting that polymeric nature of IgA antibodies contribute to cross-protection. Besides neutralizing activity, recent studies have determined the anti-influenza activities of HA specific non-neutralizing IgA antibodies against diverse influenza viruses. HA specific polymeric IgA antibodies showed no neutralizing activity but demonstrated broadly binding capacity to multiple HA subtypes including H1, H2, H5, H6, and H12 [87]. Despite the lack of neutralizing activity, IgA antibodies significantly reduced the release of most of the viruses from infected cells. Electron microscopy revealed that polymeric IgA antibodies tethered newly assembled virus particles onto cell membrane, preventing the release of the viruses into culture media. Furthermore, IgA antibodies significantly reduced the plaque size of diverse viruses in cell culture. Although in vivo protection ability of IgA antibodies was not tested, the study presents very important implications on the protective roles of mucosal IgA antibodies, suggesting that HA specific non-neutralizing IgA antibodies confer broadly protective abilities. Similar anti-influenza activity has also been reported in M2-specific IgA antibodies. Without viral neutralization, polymeric M2-specific IgA antibodies reduced plaque size and virus budding from infected cells more efficiently than monomeric IgA or IgG antibodies [88]. Those studies together show that mucosal polymeric IgA antibodies specific to influenza HA and M2 proteins have broadly protective ability by direct viral neutralization and inhibiting viral release from infected cells. The results also present possibility that IgA antibodies specific to NA may have broadly protective effects similar to antibodies to HA and M2. LAIVs induce robust IgA antibodies directed to HA, NA, and M2e, and it is likely that the antibodies cooperatively exert protective effects against a broad range of influenza viruses. Considering that mucosal IgA antibodies have ADCC activity, further studies are needed to examine the protective roles of the antibody effector functions of influenza-specific IgA antibodies [89]. 

## 5. T Cell Immune Responses

T cells can recognize endogenously- and exogenously-derived viral linear peptides generated by cleavage of viral proteins in cells. Cytotoxic CD8+ T lymphocytes (CTLs) recognize and kill virus-infected cells and help in the clearing of infected cells from the host. Harnessing T cell immune responses can overcome the limitations of UIV strategies that rely on antibodies in terms of the potency and breadth of cross-protection. Many studies have shown that the induction of cross-reactive T cell immunity is important for providing broad protection against heterologous influenza viruses and thus has been considered as an attractive strategy to construct UIVs. The importance of T cell immunity for providing broad protection and T cell-based UIV strategies have been well-described in several reviews [90,91]. This section discusses the most recent issues in T cell immunity elicited by LAIVs and their implications on developing LAIV-based UIV strategies. Early humans studies have shown that CTLs induced by LAIVs have cytotoxic activity to both homologous and heterologous influenza strains in infants and the elderly [92,93]. The detailed description of cell-mediated immunity elicited by LAIVs have been achieved by animal studies. In ferrets, LAIVs induced cross-protective CD4+ T cells and CD8+ T cells in naïve and influenza seropositive animals, providing superior protection to IIV that did not induce cell-mediated immunity [94]. In vivo T cell depletion from mice given LAIV-compromised cross-protection, demonstrated a major contribution of T cell immunity to cross-protection in the absence of neutralizing antibodies [40,95]. It has also been showed that LAIV generates lung-localized NP-specific CD4+ and CD8+ tissue resident T cells that provide long-term cross-protection in mice [96]. A LAIV generated NP- and PA-specific T cells which correlated with reduced weight loss upon influenza virus infection, diminished inflammation, and lower viral loads in the lungs compared to nonvaccinated controls in mice [97]. Human studies have also demonstrated the importance of T cell immunity to cross-protection against influenza viruses. In children, a LAIV boosted preexisting cross-reactive T cell immunity to diverse influenza strains, mainly dominated by NP-specific responses [98]. LAIVs also activated T follicular helper cells in the nasopharynx-associated lymphoid tissue that supported the production of high affinity antibodies in humans [99,100]. In addition, a LAIV was able to induce cross-reactive CD8+ T cell responses in the tonsils in children [101]. Those results together show that LAIVs can induce cross-reactive T cell responses in the respiratory tracts in humans. It is noticeable that replacement of the internal genes of LAIVs with those derived from recent circulating strains improved cross-protection [102,103]. The results suggest that LAIVs carrying the internal genes with higher similarity to those from target strains may provide better cross-protection. The currently approved types of LAIVs (A/Ann Arbor/6/60 ca and A/Leningrad/134/57 ca) are of the H2N2 subtype and, thus, may show varying degrees of differences in amino acid sequences in the internal genes with those of seasonal H1N1 and H3N2 strains and also those of H1N1 pandemics. Therefore, the use of an H1N1 LAIV, such as A/X-31 ca, that carries the H3N2 surface genes under the genetic background of the H1N1 strain may be an alternative option for providing better cross-protection against recent seasonal and pandemics influenza viruses. Of note, it has been shown that both T cells and non-neutralizing antibodies are required for robust heterosubtypic protection in mice [104]. Adoptive transfer of influenza-specific T cells into naïve mice provided little protection against lethal challenge. In addition, in vivo T cell depletion from mice with prior immunization by influenza virus infection did not abolish protection against challenge. Those results together suggest that T cells and non-neutralizing antibodies cooperate for optimal cross-protection. Analogous results were demonstrated in a study reporting that prime-boost LAIVs provided pan-influenza A protection by T cell responses and non-neutralizing ADCC-inducing antibodies in mice [40]. Thus, LAIVs are able to induce both T cell responses and antibody responses, both of which are essential for broad and potent cross-protection. It has been hypothesized that the use of IIVs interferes with the induction of virus-specific CD8+ T cells and compromises heterosubtypic cross-protection [105], which was later confirmed by animals and humans. In mice and ferrets, H3N2 IIV prevented the induction of heterosubtypic immunity to a lethal infection with the heterologous H5N1 virus and the reduction of cross-protection correlated with diminished virus-specific CD8+ T cells [106,107]. Similar diminished virus-specific CD8+ T cell responses by IIVs was also observed in humans. While unvaccinated children showed age-dependent increases of virus-specific CD8+ T cell responses, children with annual vaccination with IIVs demonstrated significantly lower frequencies and weaker trends in age-dependent increase of virus-specific CD8+ T cell responses [108]. The hampered CD8+ T cell response has been suggested as one of possible explanations for the increased risk of infection with the 2009 A/H1N1 pandemic virus in children and adults with previous annual vaccination with IIVs [109,110]. Those results give insights into the use of vaccines that can induce CD8+ T cell responses for broad protection, such as LAIVs. 

## 6. Innate Immune Responses 

In addition to adaptive immune responses, an increasing body of evidence has shown the nonspecific beneficial effects of vaccines on mortality and morbidity following infections with unrelated pathogens, which is prominent when live vaccines are used. Live virus infection stimulates the innate immune system, consisting of monocytes, macrophages, neutrophils, dendritic cells, and NK cells, consequently resulting in cytokine and chemokine production. It has been thought that innate immunity does not possess immunological memory and provides nonspecific protection and induces the same effects upon every exposure to a pathogen. However, recent data have suggested that innate immunity can also be trained and have long-term effects by epigenetic reprogramming of related genes of immune cells [111]. The nonspecific effects of vaccines have been observed in several live attenuated vaccines, including BCG vaccine (a live attenuated vaccine against tuberculosis), yellow fever vaccine, measles vaccine, and influenza live vaccine [112]. Nonspecific protection of BCG vaccine expanded not only to microbial pathogens such as *Staphylococcus aureus*, *Salmonella enteritidis*, and *Yersinia pestis*, but also to viruses such as HSV and vaccinia virus, and even to tumors in mice. Surprisingly, the protection was shown to last up to 11 months after vaccination, suggesting the exceptional breadth and duration of nonspecific effects of vaccination. Only a part of molecular mechanisms of trained innate immunity of BCG vaccine has been elucidated, including histone modification by H3K4 trimethylation resulting in the relaxation of chromatin and increased gene transcription of receptor-interacting protein kinase-2 in innate immune cells such as monocytes [113]. LAIVs have demonstrated a wide range of nonspecific protective effects on various pathogens including heterotypic influenza viruses and also unrelated microbial pathogens. A study has shown that A/X-31 ca provided heterotypic cross-protection against influenza B virus in mice, in the absence of specific antibody responses [29]. In the study, A/X-31 ca induced immediate release of IFN-α and other proinflammatory cytokines such as TNF-α, IL-6, and IL-1β, suggesting that innate immunity by a LAIV plays an important role in heterotypic protection against influenza viruses. Nonspecific protective effects of LAIVs were extended into an unrelated viral infection with respiratory syncytial virus (RSV). The replication and morbidity of RSV in mice were significantly reduced by prior immunization with A/X-31 ca, as compared to formalin-inactivated vaccine or non-vaccinated controls [114]. It was also shown that protection against RSV infection was significantly reduced in TLR3-/- TLR7-/- knock-out mice, suggesting that innate TLR signaling pathways are critical for the non-specific protection. The duration and potency of nonspecific protective effects of LAIVs warrant further investigation. In particular, whether LAIVs induce trained innate immunity that causes long-term effects by epigenetic reprogramming of immune cells would inform the design of a pan-influenza UIV that provides protection covering all types of influenza viruses. Furthermore, nonspecific protection by LAIVs is likely to provide immediate and broad protection against unrelated respiratory pathogens for which a specific vaccine is not available yet. 

## 7. Obstacles in Using LAIVs as UIVs

As discussed above, LAIV-based strategies present several benefits for developing a broad and potent UIV. The delivery of the whole set of viral antigens induces diverse antibody responses to surface and internal viral proteins. Administered via the nasal route, LAIVs stimulate mucosal immune responses that provide the first defense lines to influenza virus infection. In addition, LAIVs induce robust cell-mediated immune responses, primarily T cell immunity. Thus, the simultaneous induction of antibody responses and T cell responses against various viral proteins greatly enhance the breadth and potency of cross-protection (Table 3). In spite of these advantages, however, several issues remain to be addressed for LAIVs to serve as a reliable and practical vaccine platform for a UIV.

### 7.1. Vaccine Safety Issues

Persistent and inherent issues regarding the use of LAIVs is biosafety concerns on using live viruses as a vaccine. Although CAIVs have proven safe in humans in extensive experimental and clinical settings, the safety of other LAIVs have yet to be fully elucidated through clinical trials. Moreover, it remains possible that reversion into a virulent strain by unwanted mutations or genetic reassortment with other strains occurs in vaccinated individuals [115]. Another critical issue during the development of a UIV is vaccination-mediated enhanced viral infectivity upon infection with heterologous influenza viruses, in which vaccination may result in increased morbidity and mortality rather than protecting against the infection, termed vaccine associated enhanced respiratory disease (VAERD). The VAERD phenomena has been reported in swine model vaccinated and then challenged with heterologous influenza strains. It has been suggested that multiple factors could affect the occurrence of VAERD, including non-functional but cross-reactive HA antibodies specific to linear epitope in the HA2 subunit, vaccine adjuvant, and vaccine type [4]. 

The precise molecular mechanisms of the VAERD have not been elucidated yet, and HA stalk-based vaccines do not demonstrate the VAERD in animals and humans, suggesting that correctly-folded HA stalk antigens induce broadly protective antibodies without any harmful effects. In addition, a study has shown that HA specific antibodies that do not block receptor binding can result in antibody-dependent enhancement (ADE) of viral infectivity, highlighting the need for careful monitoring on potential harmful effects of non- or poorly neutralizing antibodies [116]. These VAERD and ADE have presented significant challenges to developing broadly protective vaccines. Several studies have shown that LAIVs do not cause VAERD while providing efficient cross-protection. A comparative study showed that a LAIV protected vaccinated pigs from heterologous and variant influenza viruses without the evidence of VAERD, whereas pigs vaccinated with IIV demonstrated significantly enhanced pneumonia following a challenge, showing typical VAERD [117]. Another study demonstrated that pigs vaccinated with IIV were not protected from heterologous pandemic H1N1 strain and demonstrated severe respiratory disease, and serum antibodies increased viral infectivity in cell culture [118]. In contrast, the study also showed that the LAIV provided cross-protection against the challenge and the serum antibodies did not result in enhanced viral infectivity, showing no evidence of VAERD. Another study also demonstrated the cross-protection of cHA-based LAIVs against the pandemic H1N1 virus in pigs, without causing VAERD [119]. A mouse study showed that prime-boost LAIVs elicit broad and potent protection against H1, H3, H5, and H7 strains and serum antibodies do not increase viral infectivity in the presence of robust HA stalk antibodies [40]. These studies together suggest that LAIVs provide cross-protection without causing VAERD and, thus, can serve as an effective and safe vaccine modality for the development of a UIV. 

### 7.2. Preexisting Immunity

Most humans acquire influenza immunity either by natural infections or vaccinations during their lifetime and preexisting immunity may impact the magnitude and hierarchy of immune responses to subsequent vaccinations. In particular, preexisting immunity can significantly affect the immune responses elicited by LAIVs that replicate in the respiratory tissues [120,121]. Early studies demonstrated that preexisting antibodies to H1N1 or H3N2 influenza strains did not interrupt the infectivity, immunogenicity, and replication of H1N1 and H3N2 LAIVs in young infants and children, suggesting that preexisting immunity does not limit the use of LAIVs in humans [122]. Furthermore, ferrets primed by trivalent LAIVs (H1N1, H3N2, and B) elicited more robust cross-reactive antibody responses following H5N1 LAIV than naive animals, presenting the beneficial effects of vaccination-induced preexisting immunity on boosting cross-reactive antibody responses by LAIVs [123]. The efficiency of boosting of HA antibodies with LAIVs showed differences between children and adults, according to their history of exposure to influenza viruses. In children those who had limited previous influenza infection or vaccination, a trivalent LAIV boosted H1 HA stalk-specific and H3 HA head-specific antibodies, supporting the use of LAIVs in children [42]. However, LAIVs rarely boosted HA antibodies to H1 and H3 HAs in adults those who already had high levels of preexisting antibodies due to repetitive exposure to diverse influenza strains. In addition, it has been shown that preexisting T cell immunity does not interfere with the replication of LAIVs and the induction of cross-reactive T cell responses following LAIVs in humans. A H5N1 LAIV boosted cross-reactive T cells directed to NP, M1, and HA in adults with the high levels of preexisting T cell immunity [124]. Likewise, a trivalent LAIV boosted preexisting T cells specific to internal NP, M1, and PB1 of heterologous H1N1 and H3N2 strains in children [98]. Thus, a common observation is that preexisting humoral immunity in adults and the elderly may limit the replication of LAIVs and decrease the immunogenicity of vaccination. Considering that antibodies are the most likely factors that inhibit the infection and replication of LAIVs [125], the use of non-human, avian subtypes such as H5, H7, or H9 LAIVs may be a helpful option to avoid preexisting immunity. Additionally, rational designs of prim-boost vaccination using non-human LAIVs and other vaccine types such as IIVs to selectively induce cross-reactive immune responses will be required for adults and the elderly. 

### 7.3. Difficulties in Precise Measurement of Correlates of Protection 

The mode of action and the breadth and potency of individual correlates of protection required for cross-protection have been well-described [126]. However, it should be noted that a LAIV induces multifaceted immune responses, including antibodies and cell-mediated immunity that collaborate on protection and, therefore, the impact of cross-protection by LAIVs, should be considered based on this point of view. As discussed above, a LAIV induces various broadly protective antibodies and T cells directed to multiple viral surface and internal antigens. Although individual correlates of protection can be quantitatively measured by a particular in vitro assay, it remains a great challenge to accurately predict the protection efficacy of a LAIV in vivo condition. For instance, it is not clear which correlate of protection between broadly neutralizing HA stalk antibodies or non-neutralizing ADCC antibodies is more effective for cross-protection. Similarly, it remains unknown whether HA stalk antibodies are more important for cross-protection than T cell immunity. It has also been shown that protection ability of broadly neutralizing HA stalk antibodies highly depend on antibody effector functions such as ADCC, ADCP, and CDC rather than neutralizing activities in vivo protection [4]. Furthermore, non-neutralizing antibodies to HA, NA, and M2e exhibit broadly protective abilities due to antibody effector functions without neutralizing activities, suggesting that antibody effector functions have critical roles in broad protection in vivo. Importantly, protective levels of correlates of protection have not been clearly defined, providing additional difficulty in predicting the protective efficacy of a LAIV. While HI antibody titers greater than 40 in serum has been considered as protective, no defined protective titers have been clearly suggested for other correlates. In addition, protective level of each correlate likely varies substantially depending on the antigenic distance between a vaccine and a target virus. Antigenic differences between a vaccine and a target virus will affect the antibody affinity to viral antigens, thus influencing the protective level of each correlate required for protection. 

### 7.4. Population-Level Implications

Several studies presented mathematical modeling analysis to demonstrate the benefits of using UIV for controlling seasonal influenza and pandemic influenza at a population level. It has been generally expected that although UIVs do not block viral infection against a given strain, they present superior abilities to reduce the epidemic size, pandemic size, and the antigenic evolution of the viruses over strain-matched vaccines [127,128]. However, it should be noted that this anticipation by mathematical modeling was drawn under the assumption that UIVs act uniformly against all influenza strains, which is very unlikely in a real situation. Another recent study suggested the potential unintended negative effects of the use of UIVs at a population level. The study used a mathematical modelling and predicted adverse consequences of wide use of seasonal UIV program [129]. Annual vaccination with UIVs may reduce the opportunity to obtain the ‘infection-induced immunity’ by seasonal epidemic strains in the population. This may cause more severe impacts when a vaccine escape variant or a novel pandemic strain against which the UIV cannot protect is newly introduced to the population. It was also anticipated that the risk could be reduced by achieving optimal protection efficacy and sufficient vaccination coverage in the population through the combined use of strain-matched vaccines with UIVs or through using UIVs carrying multiple antigenic targets. This prediction points out the potential negative effects of using UIVs that depend heavily on antibody responses to a single conserved epitope, since the reduction of epitope-based protection through genetic mutations cannot be sufficiently compensated by other immune responses. LAIVs, mimicking natural infection, induce the ‘infection-induced immunity’ and deliver multiple universal targets, thus presenting advantages for providing optimal and robust protection due to multifaceted immune correlates. Prime vaccination with a strain-matched LAIV followed by boost vaccination with a heterosubtypic LAIV under the same genetic background would elicit both strain-specific and broad heterosubtypic protection. In the event of emergence of vaccine variant strain or a novel pandemic strain, LAIVs likely remain protective due to multiple immune responses cross-protective to the viruses. In the future, it is likely that strain-matched vaccines and UIVs may play complementary roles to guarantee both potency and breadth of protection. Those modeling studies consistently show that high protection efficacy and vaccination coverage are the prerequisite for the beneficial effects of a UIV. LAIVs are delivered by nasal spray and the ease of administration presents a better option for mass influenza vaccination programs than parental vaccines. 

## 8. Rational Designs of UIV Strategies Using LAIVs

This section suggests several rational vaccination strategies using LAIVs for the development of broad, potent, and safe UIVs. Provided that strain-specific neutralizing antibody responses toward highly variable HA head do not provide uniform broad protection across influenza viruses, UIV strategies based on LAIVs do not have to include particular sets of HA and NA surface antigens, presenting considerable flexibility for rational designs using LAIVs. Considering that the HA gene of a cold-adapted master donor strain also has attenuating mutations [130], the use of cold-adapted donor strains without replacing surface genes with those originated from wild-type viruses may result in the increased safety of a UIV. Prime-boost vaccination with LAIVs each carrying different subtypes of HA and NA under the same backbone strain may result in boosted cross-reactive T cell responses directed to internal viral proteins. Consistent with this speculation, prime vaccination with H1N1 LAIV followed by boost vaccination with H5N1 LAIV demonstrated more potent cross-protection against both HA group 1 and group 2 influenza A viruses than prime-boost vaccination with H1N1 LAIVs in a mouse model [40]. Whether prime-boost vaccination with group 1 and group 2 LAIVs would provide broader protection merits further investigation. Although influenza B viruses are considered secondary targets for developing a UIV, similar approaches can be extended into pan-influenza B UIVs. Although there are two lineages in influenza B viruses circulating in humans, they are not classified into distinct subtypes and show much lower genetic variability than influenza A viruses. Therefore, the development of UIVs against influenza B viruses may be easily attainable compared to those for influenza A viruses. Evaluation of the potency and breadth of type B LAIVs against both lineages of influenza B viruses would be needed. One or two times of vaccination with a bivalent vaccine formulation containing both type A LAIV and type B LAIV could be suggested for the development of pan-influenza UIV against all influenza viruses. As described above, in addition to cold-adapted LAIVs, several approaches have been developed to construct novel LAIVs, including NS1 truncation, host protease-mediated cleavage of influenza proteins, and miRNA-mediated gene silencing (Table 1). Demonstrating various profiles of potency and breadth, those LAIVs can also be used for constructing better UIVs. Furthermore, it would be possible to generate double attenuated LAIVs by combination of those attenuation tools for increased safety. For instance, a H1N1 LAIV containing both elastase cleavage site in the HA and NS1 truncation showed full attenuation but was still immunogenic providing potent and broad protection against homosubtypic H1N1 and heterosubtypic H3N2 challenges in pigs [15]. 

## 9. Conclusions

The discovery of broadly neutralizing antibodies against influenza viruses does not translate directly into the development of broadly protective influenza vaccines. While the antibodies present promising strategies for broad therapeutics against the viruses, the induction of such antibodies by vaccination may be considerably difficult. It has been shown that vaccination induce many different polyclonal antibodies directed to HA stalk, each of which differs in binding breadth, protection potency, and the mode of protection. These points may also hold true for antibodies directed to the other viral antigens, such as NA, M2e, and internal proteins. Furthermore, a concern has also been raised on the unexpected induction of non-functional and harmful antibodies that enhance viral infectivity rather than protecting against the virus, as shown by antibodies to the linear epitope of HA2 region. The delivery of correctly folded antigens can overcome VAERD problem by HA stalk antibodies. However, the use of correctly folded antigens necessarily leads to the induction of polyclonal antibodies that have different characteristics. In addition to HA stalk antibodies, it has been shown that the expression of corrected folded NA antigens is essential for eliciting broadly protective NA antibodies by vaccination. In addition, the induction of both antibody responses and T cell responses are required for providing sufficiently broad and potent cross-protection. For this purpose, prime-boost vaccine regimens using different vaccine types are preferred, mostly in combination of IIVs and LAIVs. Mucosal immunity should also be noted as a critical aspect of LAIVs since mucosal antibodies directed to influenza virus surface proteins have been shown to demonstrate broad protective activities. Several animal studies have shown that LAIVs alone can provide exceptionally broad and potent protection by the induction of multiple immune correlates without neutralizing antibodies, suggesting that LAIVs can serve as a standalone vaccine strategy for a UIV. However, several limitations of LAIVs should also be noted, such as low vaccine effectiveness in adults and the elderly, poorly characterized immune correlates of cross-protection, and the lack of controlled human challenge studies. For clinical relevance, considerable efforts are needed to design rational approaches that are able to efficiently induce cross-protective immune responses by LAIVs. 

## Figures and Tables

**Figure 1 vaccines-09-00353-f001:**
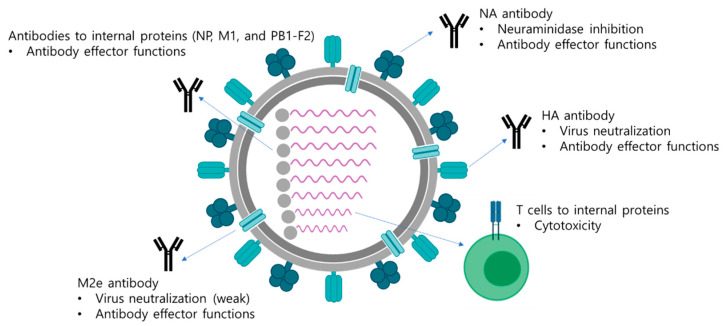
Multifaceted immune responses induced by a LAIV. A LAIV mimics a natural infection and induces multifaceted immune responses including antibody responses and T cell responses. Antibodies directed to surface antigens, HA, NA, and M2e neutralize the virus and also mediate antibody effector functions, such as ADCC, ADCP, and CDC, to eliminate virus-infected cells. The internal viral proteins are also target antigens for generation of antibodies with effector functions. Whether the effector functions of antibodies directed to the internal proteins have protective roles has not been clearly elucidated yet. In addition, cytotoxic T cells are directed to the internal viral proteins, which mediate the elimination of virus-infected cells via direct contact of cytokine production.

**Table 1 vaccines-09-00353-t001:** Strategies for construction of LAIVs.

Strategy	Mechanism of Attenuation
Cold-adaptation	Genetic mutations accumulated during cold-adaptation result in decreased viral replication at body temperature.
Deletion or truncation of NS1	The lack of interferon antagonist NS1 protein results in decreased viral replication in infected cells.
Deletion of M2 ion channel	M2-deficient influenza viruses replicates only in cells expressing M2 proteins but are highly restricted in normal cells.
Caspase-dependent cleavage of viral proteins	Cleavage of viral proteins by caspases activated during apoptosis of infected cells results in decreased viral replication.
Modification of HA cleavage site	The mutant viruses carrying elastase cleavage site in HA undergo restricted replication because of the absence of appropriate protease.
miRNA-mediated gene silencing	The viral genes carrying miRNA-targeted region are degraded in infected cells.
Codon deoptimization	Codon deoptimization results in downregulation of viral protein synthesis in infected cells.
Engineering of splicing elements	Modification of splicing elements in viral genes results in decreased production of the proteins in infected cells.

**Table 2 vaccines-09-00353-t002:** Cross-protective immune responses elicited by LAIVs.

Immune Responses	Mechanism of Attenuation
HA stalk antibodies	Viral neutralization Antibody effector functions
NA antibodies	Neuraminidase inhibition Antibody effector functions
M2e antibodies	Viral neutralization Antibody effector functions
Antibodies to internal proteins	Antibody effector functions
T cell response	Cytotoxicity to virus-infected cells
Mucosal immunity	Viral neutralization Non-neutralizing activity
Innate immunity	Non-specific effects

**Table 3 vaccines-09-00353-t003:** Benefits and obstacles in using LAIVs as UIVs.

Benefits	Obstacles
Delivery of whole set of viral antigens including surface and internal proteins	Low immunogenicity in adults and the elderly due to preexisting immunity
Simultaneous induction of humoral and cell-mediated immune responses	Difficulties in precise evaluation of correlates of protection in vivo condition
Robust immunogenicity in seronegative population and children	Possible reversion into a virulent strain by mutations or genetic reassortment with other strains
Acquirement of infection-induced immunity	Over attenuation may compromise the productivity and immunogenicity of LAIVs
Ease of administration by nasal spray

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
