# Peer review of "Immune Responses Elicited by Live Attenuated Influenza Vaccines as Correlates of Universal Protection against Influenza Viruses"

_vaccines, 2021, doi:10.3390/vaccines9040353_

Round 1

Reviewer 1 Report

This review summarized the immune responses induced by live attenuated influenza vaccines (LAIVs), emphasizing the breadth and the potency of individual immune correlates. Generally, the manuscript was well organized and written, providing insights into the development of a universal influenza vaccine.

Major concerns:

  1. Informative and insightful figures and tables created by the authors are required.
  2. The potential biosafety concerns and practical applicability prospects regarding the LAIVs should be highlighted.
  3. The manuscript needs to be improved by moderate language editing.

Author Response

This review summarized the immune responses induced by live attenuated influenza vaccines (LAIVs), emphasizing the breadth and the potency of individual immune correlates. Generally, the manuscript was well organized and written, providing insights into the development of a universal influenza vaccine.

Major concerns:

1.Informative and insightful figures and tables created by the authors are required.

Response: The figure and tables were slightly modified to include more detailed information.

2. The potential biosafety concerns and practical applicability prospects regarding the LAIVs should be highlighted.

Response: Statements describing the potential biosafety issues of LAIVs were added with relevant reference in 7.1 subsection (page 12, lines 559-564).

Persistent and inherent issues regarding the use of LAIVs is biosafety concerns on using live viruses as a vaccine. Although CAIVs have proven safe in humans in extensive experimental and clinical settings, the safety of other LAIVs have yet to be fully elucidated through clinical trials. Moreover, it remains possible that reversion into a virulent strain by unwanted mutations or genetic reassortment with other strains occurs in vaccinated individuals [115].

3.The manuscript needs to be improved by moderate language editing.

Response: The manuscript was sent to and revised by a professional English editing service

Reviewer 2 Report

General comments:

The authors have made a comprehensive review of the literature on immune responses to LAIVs and also discussed the limitations of LAIVs as UIVs. However, they did not adequately address how the different immune responses could be correlated to universal protection or cross-protection against influenza viruses.  As discussed in Section 7.3, this is a challenging task and is a work in progress. Nevertheless, this makes the review title somewhat misleading.  

Additional comments:

  1. Is it known for how long do HA, NA antibodies to influenza viruses last, following (i) natural infection, and vaccination with (ii) LAIVs and (iii) IIVs? Please provide some information on this point. Could waning or short-lived immunity be also partly responsible for poor protection?
  2. Section 2.1, CAIVs: it will be useful to provide more details of the genetic changes or specific characteristics responsible for the attenuated phenotypes of the 5 strains of CAIVs used commercially.
  3. Pg. 4, line 141-143: can the authors provide the summary of the outcomes of the follow-up studies (references 11, 12). This will be useful to readers to access the utility of the attenuated influenza virus from reference 10 as a vaccine candidate.
  4. Pg. 6, lines 214-222: it is not clearly explained how the antibody effector function, such as interaction with Fc receptors, can yield broad protection. Please elaborate on the immunological response(s) from Fc-FcR interaction that can lead to broad protection against different influenza viruses.
  5. Pg. 6, ferret studies in reference 37 and 41: No HA stalk antibodies were generated after three doses of LAIVs in ref. 37, whilst one or two doses of LAIVs induced HA stalk antibodies. Please highlight the difference between these studies.
  6. Pg. 6, line 249-250 & Pg. 7, lines 259-260 : it is probably important to induce HA stalk antibodies with multiple types of effector functions that collectively induce broad protection. Please comment.
  7. Section 5, refs 88-97: do these references specify the viral protein(s)/epitopes that are responsible for inducing the T cell responses? Please provide the information if available.
  8. Section 7.1 and Pg. 14, lines 635-641: whilst no HA stalk-based vaccine has not demonstrated VARED, Winarski K.L. et al. (PNAS 2019) described ADE with two different functional MAbs that destabilized HA stem domain increased influenza virus fusion kinetics, and led to enhanced lung pathology and ERD in a dose-dependent manner in a mice model.The risk of ADE from non-or poorly neutralising antibodies should not be overlooked.
  9. Pg. 13, lines 592-5, what is meant by the phrase “LAIVs infect and replicate for optimal immunogenicity…”? Regarding the following phrase “… which are mechanistically the same to the viral infection”: shouldn’t the repertoire and duration of immune response differ between LAIVs and wildtype viruses? Please rephrase as needed.
  10. Pg. 696-8 and Pg. 708-710: how does priming with H1N1 LAIV followed by boosting with H5N1 LAIV differ from two or three doses of bi-, tri- or tetra valent LAIV comprising groups 1, 2 IAV and/or IBV lineages, in terms of inducing broad protection? Why is this dosing regime not adopted?
  11. Section 9, Conclusions: the authors have not touched upon the importance of mucosal immunity and its relevance for assessing vaccine efficacy. This is a critical aspect of LAIVs and should be emphasized.

Minor comments:

  1. Pg. 2, line 46: change typo-error “stain” to “strain”.
  2. Pg. 2, line 75: please clarify if the reference to “HA group 1 and 2” is intended only for IAV subtypes?
  3. Pg. 2, line 87: please change phrase “ making the virus to less replicate” to “ making the virus less replicative”.
  4. Pg. 4, line 143-146: similar comment for refs 13-15 as point 5.
  5. Section 2.2: another factor to consider is whether it is feasible to produce the various types of LAIVs in sufficiently large quantities for commercialization.
  6. Pg. 7, line 286: suggest to change “constant” to “consistent”.
  7. Pg. 7, line 296: change to “ NA is the second most abundant”.
  8. Pg. 8, lines 337-8: change to “one of the benefits’
  9. Pg. 8, line 353: change to “Given the highly”.
  10. Pg. 9, lines 364-5: suggest to change to “NP antibodies had low but detectable CDC activity [65], showing results vary depending….”.
  11. Pg. 9, line 366: change to “NP and M1 are…”
  12. Pg. 10, line 443: suggest to change to “ help in the clearing of”.
  13. Pg. 10, line 449: suggest to delete “a”.
  14. Pg. 10, line 457: suggest to change” than” to “to”
  15. Pg. 11, line 511: suggest to “since”.
  16. Pg. 12, lines 524-3: please specify which cells undergo the histone modification.
  17. Pg. 12, line 559: suggest to change to “results in increased morbidity and mortality”.
  18. Pg. 13, Section 7.2: statements in lines 592-595.
  19. Pg. 15, line 699: change “influenza A viruses” or “IAVs”.
  20. Pg. 16, line 724: change to “may be considerably”.
  21. Pg. 16, line 728: change to “Furthermore”.

Author Response

The authors have made a comprehensive review of the literature on immune responses to LAIVs and also discussed the limitations of LAIVs as UIVs. However, they did not adequately address how the different immune responses could be correlated to universal protection or cross-protection against influenza viruses. As discussed in Section 7.3, this is a challenging task and is a work in progress. Nevertheless, this makes the review title somewhat misleading.

Additional comments:

1.Is it known for how long do HA, NA antibodies to influenza viruses last, following (i) natural infection, and vaccination with (ii) LAIVs and (iii) IIVs? Please provide some information on this point. Could waning or short-lived immunity be also partly responsible for poor protection?

Response: It is reported that antibody responses after influenza infection or vaccination (LAIV or IIV) last for more than six months to 1 year, which is sufficient period of time for durable protection against each epidemic strain (J Infect Dis 2015;211:1541, Am J Respir Crit Care 2015;191:325, Vaccines 2012;30:5533). Therefore, it does not seem that short-live antibody responses are responsible for poor protection of any vaccines.

2.Section 2.1, CAIVs: it will be useful to provide more details of the genetic changes or specific characteristics responsible for the attenuated phenotypes of the 5 strains of CAIVs used commercially.

Response: It was shown that the genetic mutations accumulated in the polymerase genes and NP gene during cold-adaptation are responsible for attenuated phenotype of CAIVs. A sentence describing this point was added in the text in relevant references (page 2, lines 84-89).

Genetic and phenotypic analysis have revealed that mutations in the polymerase genes and the NP gene are crucial for the expression of the ca, ts, and att phenotypes [5-8].

3.Pg. 4, line 141-143: can the authors provide the summary of the outcomes of the follow-up studies (references 11, 12). This will be useful to readers to access the utility of the attenuated influenza virus from reference 10 as a vaccine candidate.

Response: Sentences describing the results of follow-up studies were added in the text (page 4, lines 142-144).

A double attenuated LAIV with elastase-susceptible HA cleavage site and shortened NS1 protein demonstrated increased safety but was still immunogenic in swine model [15]. Similar strategy could be extended to LAIV against influenza B virus [16].

4.Pg. 6, lines 214-222: it is not clearly explained how the antibody effector function, such as interaction with Fc receptors, can yield broad protection. Please elaborate on the immunological response(s) from Fc-FcR interaction that can lead to broad protection against different influenza viruses.

Response: Please note that antibody effector function itself does not broaden the protection. Antibodies directed to conserved regions such as HA stalk and M2e have cross-protective activities via direct neutralizing activity of antibody effector function.

5.Pg. 6, ferret studies in reference 37 and 41: No HA stalk antibodies were generated after three doses of LAIVs in ref. 37, whilst one or two doses of LAIVs induced HA stalk antibodies. Please highlight the difference between these studies.

Response: In the former study, three LAIVs carry full-length but non-chimeric HAs and thus the HA stalk of each LAIV is different to each other. The latter study used chimeric Has carrying the same HA stalks. This may explain the difference between the two studies. Sentences were rephrased to clearly indicate this point (page 6, lines 233-235).

However, a ferret study showed that three doses of LAIVs carrying full-length and non-chimeric HAs including H5, H8, and H9 rarely induced HA stalk antibodies, suggesting that HA stalk antibodies were poorly induced by boosting with different HA stalks [41].

6.Pg. 6, line 249-250 & Pg. 7, lines 259-260 : it is probably important to induce HA stalk antibodies with multiple types of effector functions that collectively induce broad protection. Please comment.

Response: The reference 43 in the revised m/s showed only ADCC results but not the other antibody effector functions. And the reference 45 in the revised m/s showed only HA stalk antibody titers but not any antibody effector functions. Therefore, it would be appropriate to describe only presented data.      

7.Section 5, refs 88-97: do these references specify the viral protein(s)/epitopes that are responsible for inducing the T cell responses? Please provide the information if available.

Response: Reference 96, 97, 98 specify viral epitopes that are responsible for inducting T cell responses. Sentences were revised to include this information (page 10, lines 460-468).

It has also been showed that LAIV generates lung-localized NP-specific CD4+ and CD8 + tissue resident T cells that provide long-term cross-protection in mice [96]. A LAIV generated NP and PA specific T cells which correlated with reduced weight loss upon influenza virus infection, diminished inflammation, and lower viral loads in the lungs compared to nonvaccinated controls in mice [97]. Human studies have also demonstrated the importance of T cell immunity to cross-protection against influenza viruses. In children, A LAIV boosted preexisting cross-reactive T cell immunity to diverse influenza strains, mainly dominated by NP-specific responses [98].

8.Section 7.1 and Pg. 14, lines 635-641: whilst no HA stalk-based vaccine has not demonstrated VARED, Winarski K.L. et al. (PNAS 2019) described ADE with two different functional MAbs that destabilized HA stem domain increased influenza virus fusion kinetics, and led to enhanced lung pathology and ERD in a dose-dependent manner in a mice model. The risk of ADE from non-or poorly neutralising antibodies should not be overlooked.

Response: Following the reviewer’s suggestion, a sentence describing the potential ADE of non- or poorly neutralizing antibodies was added to the text (page 13, lines 577-580).

In addition, a study has shown that HA specific antibodies that do not block receptor binding can result in antibody-dependent enhancement (ADE) of viral infectivity, highlighting the need for careful monitoring on potential harmful effects of non- or poorly neutralizing antibodies [116].

9.Pg. 13, lines 592-5, what is meant by the phrase “LAIVs infect and replicate for optimal immunogenicity…”? Regarding the following phrase “… which are mechanistically the same to the viral infection”: shouldn’t the repertoire and duration of immune response differ between LAIVs and wildtype viruses? Please rephrase as needed.

Response: The sentence was rephrased as follows for clarity (page 13, lines 601-603).

In particular, preexisting immunity can significantly affect the immune responses elicited by LAIVs that replicate in the respiratory tissues [120,121].

10.Pg. 696-8 and Pg. 708-710: how does priming with H1N1 LAIV followed by boosting with H5N1 LAIV differ from two or three doses of bi-, tri- or tetra valent LAIV comprising groups 1, 2 IAV and/or IBV lineages, in terms of inducing broad protection? Why is this dosing regime not adopted?

Response: The reviewer pointed out the use of multivalent formulation with different LAIVs for broad protection. Please note that the text already included a sentence describing very similar point with that of the reviewer (page 15, lines 714-716).

One or two times of vaccination with a bivalent vaccine formulation containing both type A LAIV and type B LAIV could be suggested for the development of pan-influenza UIV against all influenza viruses.

11.Section 9, Conclusions: the authors have not touched upon the importance of mucosal immunity and its relevance for assessing vaccine efficacy. This is a critical aspect of LAIVs and should be emphasized.

Response: As suggested by the reviewer, a sentence describing the importance of mucosal immunity was added to the text (page 16, lines 743-746).

Mucosal immunity should also be noted as a critical aspect of LAIVs since mucosal antibodies directed to influenza virus surface proteins have been shown to demonstrate broad protective activities.

Minor comments:

1.Pg. 2, line 46: change typo-error “stain” to “strain”.

Response: “stain” was corrected to “strain” (page 2, lines 45).

2.Pg. 2, line 75: please clarify if the reference to “HA group 1 and 2” is intended only for IAV subtypes?

Response: ‘influenza viruses’ was changed into ‘influenza A viruses’ for clarity (page 2, lines 73).

3.Pg. 2, line 87: please change phrase “ making the virus to less replicate” to “ making the virus less replicative”.

Response: The phrase was changed as suggested by the reviewer (page 2, lines 85).

4.Pg. 4, line 143-146: similar comment for refs 13-15 as point 5.

Response: Follow-up studies on miR-mediated attenuated LAIVs were shortly discussed as follows (page 4, lines 146-148).

miR-21- or miR-192 targeted influenza viruses were highly attenuated in susceptible cells and mice but provided robust homologous and heterologous protection in mice [18,19].

5.Section 2.2: another factor to consider is whether it is feasible to produce the various types of LAIVs in sufficiently large quantities for commercialization.

Response: A sentence describing the vaccine productivity was added to the text (page 4, lines 153-154).

It should also be noted that over attenuation of the virus may limit vaccine productivity and also the immunogenicity of LAIVs

6.Pg. 7, line 286: suggest to change “constant” to “consistent”.

‘Constant’ was corrected to ‘consistent’(page 7, lines 288).

7.Pg. 7, line 296: change to “ NA is the second most abundant”.

Response: ‘most’ is inserted to the sentence (page 7, lines 298).

8.Pg. 8, lines 337-8: change to “one of the benefits’

Response: ‘the’ is inserted to the sentence (page 8, lines 339).

9.Pg. 8, line 353: change to “Given the highly”.

Response: ‘the’ is inserted to the sentence (page 8, lines 355).

10.Pg. 9, lines 364-5: suggest to change to “NP antibodies had low but detectable CDC activity [65], showing results vary depending….”.

Response: The sentence was revised for clarity as follows (page 9, lines 367).

Human NP antibodies demonstrated no detectable CDC activity [68], whereas murine NP antibodies low but detectable CDC activity [69], showing contrasting results depending on species or experimental settings.

11.Pg. 9, line 366: change to “NP and M1 are…”

Response: The sentence was corrected (page 9, lines 368).

12.Pg. 10, line 443: suggest to change to “ help in the clearing of”.

Response: The sentence was corrected as suggested by the reviewer (page 9, lines 444).

13.Pg. 10, line 449: suggest to delete “a”.

Response: The sentence was corrected (page 10, 449).

14.Pg. 10, line 457: suggest to change” than” to “to”

Response: The sentence was corrected (page 10, lines 457).

15.Pg. 11, line 511: suggest to “since”.

Response: ‘since’ was deleted in the sentence (page 11, lines 512).

16.Pg. 12, lines 524-3: please specify which cells undergo the histone modification.

Response: The sentence was revised to indicate the cell type that undergoes histone modification (page 12, lines 526-527).

Only a part of molecular mechanisms of trained innate immunity of BCG vaccine has been elucidated, including histone modification by H3K4 trimethylation resulting in the relaxation of chromatin and increased gene transcription of receptor-interacting protein kinase-2 in innate immune cells such as monocytes [113].

17.Pg. 12, line 559: suggest to change to “results in increased morbidity and mortality”.

Response: ‘the’ was deleted (page 12, lines 566).

18.Pg. 13, Section 7.2: statements in lines 592-595.

Response: The sentence was rephrased for clarity. Please see the response to major comment 9 (page 13, lines 601-603).

In particular, preexisting immunity can significantly affect the immune responses elicited by LAIVs that replicate in the respiratory tissues [120,121]. 

19.Pg. 15, line 699: change “influenza A viruses” or “IAVs”.

Response: ‘influenza viruses’ was changed to ‘influenza A viruses’ (page 15, lines 704-705).

20.Pg. 16, line 724: change to “may be considerably”.

Response: The sentence was corrected (page 16, lines 729).

21.Pg. 16, line 728: change to “Furthermore”.

Response: ‘Further’ was changed to ‘Furthermore’ (page 16, lines 733).

Reviewer 3 Report

The review article submitted by Yo Han Jang et al provides us an overview of multifaceted immune responses induced by live attenuated influenza vaccines and correlates for influenza virus protection. The development of universal influenza vaccines remains to be a challenging task and many strategies for universal vaccine development have been assessed in recent years with each possessing unique pros and cons.  The review is comprehensive and up-to-date, and the information covered in this article is accurate. Overall, it is a good review and well written, and will be a good addition to the influenza research community.

Minor comments
1. Line 94 “serve” should be “served”.

2. Line 420, “H” should be “H1”.

Author Response

The review article submitted by Yo Han Jang et al provides us an overview of multifaceted immune responses induced by live attenuated influenza vaccines and correlates for influenza virus protection. The development of universal influenza vaccines remains to be a challenging task and many strategies for universal vaccine development have been assessed in recent years with each possessing unique pros and cons. The review is comprehensive and up-to-date, and the information covered in this article is accurate. Overall, it is a good review and well written, and will be a good addition to the influenza research community.

Minor comments

  1. Line 94 “serve” should be “served”.

Response: The sentence was corrected (page 2, lines 94).

  1. Line 420, “H” should be “H1”.

Response: The sentence was corrected (page 10, lines 420).